# Single planar photonic chip with tailored angular transmission for multiple-order analog spatial differentiator

Yang Liu [1], Mingchuan Huang [1], Qiankun Chen [1] & Douguo Zhang [1,2] ✉

Analog spatial differentiation is used to realize edge-based enhancement, which plays an important role in data compression, microscopy, and computer vision applications. Here, a planar chip made from dielectric multilayers is proposed to operate as both first- and second-order spatial differentiator without any need to change the structural parameters. Third- and fourth-order differentiations that have never been realized before, are also experimentally demonstrated with this chip. A theoretical analysis is proposed to explain the experimental results, which furtherly reveals that more differentiations can be achieved. Taking advantages of its differentiation capability, when this chip is incorporated into conventional imaging systems as a substrate, it enhances the edges of features in optical amplitude and phase images, thus expanding the functions of standard microscopes. This planar chip offers the advantages of a thin form factor and a multifunctional wave-based analogue computing ability, which will bring opportunities in optical imaging and computing.

Currently, there is significant interest in optical analog signal processing because of the rapid increase in demand for large-scale, real-time data processing in the big data era, and the major advances being made in nanophotonics are leading to opportunities in this field. Although digital signal processing can offer great versatility, it suffers from problems with low processing speeds, high power consumption, and high complexity caused by the use of costly analogue-to-digital converters[1], and these problems become obvious during the investigation of nonrepetitive and rare phenomena (e.g., nonlinear dynamics)[2]. These restrictions can be removed by using optical analog signal processing[2–5], because it has an intrinsically parallel nature that offers high-speed operation and low power consumption. The conventional bulky lenses that are used in traditional optical signal processing and Fourier optics[6] have now been replaced with nanophotonic materials, such as the compactible and power-efficient ultrathin devices that have been fabricated based on metasurfaces (or metamaterials)[7–10], photonic crystals[11,12], plasmonic structures[13,14], spin Hall effect[15,16] and topological photonics[17].

The optical analog spatial differentiator is one of these devices. This differentiator can enable massively parallel processing of the edges detected from an entire image[18], which has been shown to have important applications in machine and computer vision[19], medical or biological imaging operation[20,21] and autonomous vehicles[22,23]. However, most optical analog spatial differentiators based on nanophotonic materials can perform only one mathematical operation, producing either the first-order or the second-order derivative[8,9,11–13,17,24–27]. Complex materials and fabrication processes are also required for these devices, which contain artificially engineered structures.

In this work, a planar photonic chip composed of an all-dielectric multilayer structure is proposed and its nonlocality (or named as angular-dependent transmission) is tailored to enable signal manipulation in the momentum domain[28,29]. It can be used to perform various types of mathematical operations on optical signals, such as the first-, second-, third-, fourth- and even higher-order spatial derivative operations without altering the chip's structural parameters. This chip can be fabricated on a large scale using standard deposition

[1]Advanced Laser Technology Laboratory of Anhui Province, Department of Optics and Optical Engineering, University of Science and Technology of China, Hefei, Anhui 230026, China. [2]Hefei National Laboratory, University of Science and Technology of China, Hefei 230088, China. ✉e-mail: dgzhang@ustc.edu.cn

techniques. Edge detection is realized for both amplitude and phase objects when this chip is integrated into the optical path of a commercial optical microscope.

## Results

### Principle of the multiple-order analog spatial differentiation

The relationship between the input electric field $\mathbf{E_{in}}(x,y)$ and the output electric field $\mathbf{E_{out}}(x,y)$ for an optical device in Cartesian coordinate system can be described as follows:

$$\mathbf{E_{in}}(x,y) = \mathbf{e_{in}} \iint A(k_x, k_y) \exp(ik_x x + ik_y y) dk_x dk_y \quad (1)$$

$$\mathbf{E_{out}}(x,y) = \mathbf{e_{out}} \iint t(k_x, k_y) A(k_x, k_y) \exp(ik_x x + ik_y y) dk_x dk_y \quad (2)$$

where $A(k_x, k_y)$ is the amplitude of the field, $k_x, k_y$ are the $x$- and $y$-components of the in-plane wavenumber, respectively, and $t(k_x, k_y)$ represents the optical transfer function (OTF)[30]. $\mathbf{e_{in}}$ and $\mathbf{e_{out}}$ represent the polarization states of input and output fields in real space, which are controlled by the polarizers elements used in the experiments.

To realize first-order spatial differentiation along $x$-direction, the condition $\mathbf{E_{out}} \propto \partial \mathbf{E_{in}}/\partial x$ should be satisfied, and then $t(k_x, k_y) \propto k_x$. In the same way, to achieve higher order one-dimensional differential operations, $t(k_x, k_y)$ should have the forms of $t(k_x, k_y) \propto k_x^2, k_x^3, k_x^4, \cdots\cdots$. In the polar coordinate system, to achieve multi-order radial differential operations ($\mathbf{E_{out}} \propto \partial^n \mathbf{E_{in}}/\partial r^n$) which enable two-dimensional processing, the optical transfer function should satisfy the forms of $t(k_r, \varphi) \propto k_r^1, k_r^2, k_r^3, k_r^4, \cdots\cdots$, where $k_r = \sqrt{k_x^2 + k_y^2}$ is the in-plane wavevector along the radial direction.

In this work, a planar photonic chip made from a dielectric multilayer structure is designed to realize these first-, second-, and even higher order spatial differentiation operations on the same chip, as illustrated in Fig. 1a. The chip is composed of alternating layers of $Si_3N_4$ and $SiO_2$. There are 40 of these layers in total. The thicknesses of the $Si_3N_4$ and $SiO_2$ layers are 56 nm and 80 nm, respectively. Further details of the structural parameters of this chip are provided in the supplementary information. The transmission band diagram of the multilayer structure is calculated using the transfer matrix method (TMM)[31], as shown in Fig. 1b, where the $p$- and $s$-polarization are plotted on the right and left, respectively. The dark blue region corresponds to the forbidden band, and the red dashed line indicates the frequency of 467 THz ($\lambda = 643$ nm), at which the transmissivity satisfied the function $t = t_0 + t_2 k_r^2$ as illustrated in Fig. S1b in the supplementary information, where $t_0$ and $t_2$ are fitting coefficients. Note that $t_0$ can be regarded as the direct transmission component, which can be reduced to zero when an analyzer is used. The direct transmission component is not related to the wavenumber of the incident light, and will form the background of the output optical field.

The angular-dependent transmission image of this multilayer structure at $\lambda = 643$ nm was calculated as shown in Fig. 1c. The gradient color arrows represent the polarization orientation of the output electric field due to the difference between transmission coefficients $t_s$ and $t_p$. The figure shows that the polarization orientation of the output

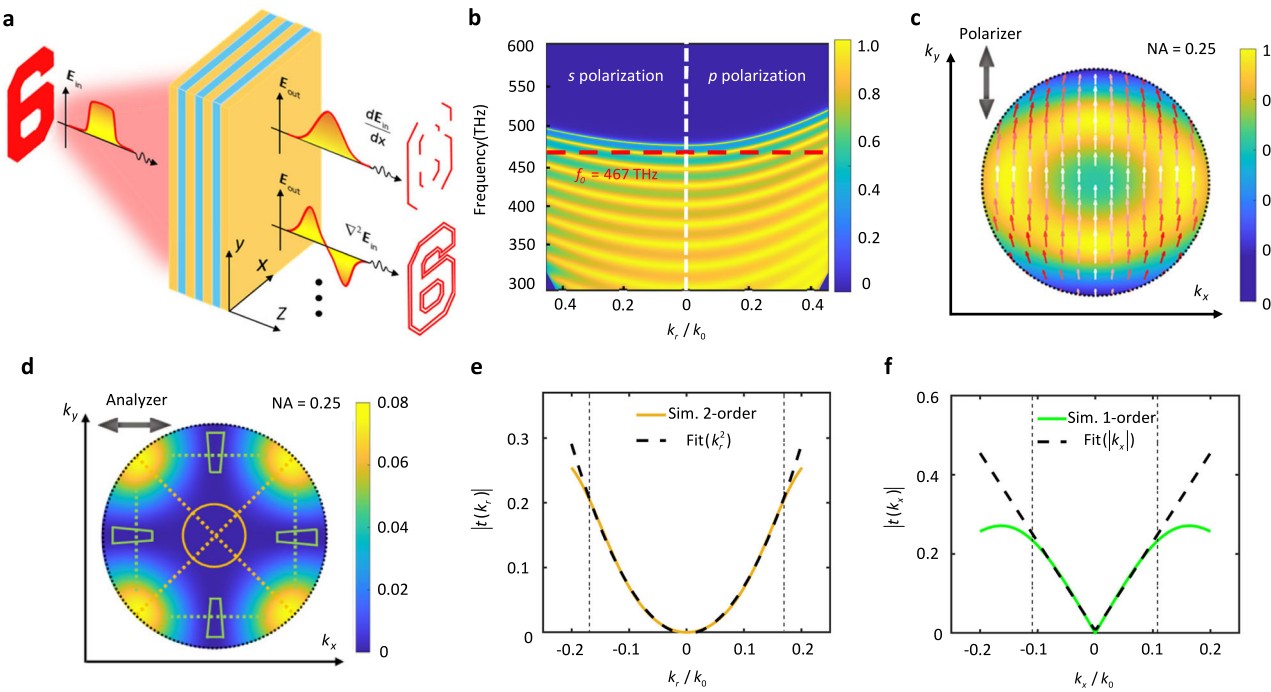

**Fig. 1 | Planar photonic chip for spatial differentiation. a** Schematic of a photonic chip acting as a spatial differentiator that transforms an image into its first-, second- and even higher order derivative. The chip is made from a well-designed dielectric multilayer structure. **b** Simulated color-coded transmission coefficient amplitude as a function of frequency and in-plane wavenumber for the $s$- and $p$-polarizations. **c** Simulated transmission BFP image through the dielectric multilayer at an incident wavelength of 643 nm. The double arrow-headed black line at the top left corner indicates the polarization orientation of the incident light. The colored one-way arrows indicate the polarization direction of transmitted light in the momentum space, the polarization deflection increases as the color change from white to red. **d** Simulated transmission BFP image when an analyzer is placed after the dielectric multilayer. The polarization orientation of the analyzer lies perpendicular to the incident polarization orientation, as indicated by the double arrow-headed line. The green trapezoidal regions are designed to perform first-order differentiation, and the yellow circular region for second-order differentiation. The green dotted lines and the yellow dotted lines indicate the directions of first-order differentiation and second-order differentiation, respectively. **e** Simulated optical transfer function $|t(k_r)|$ for the second-order differentiation at $\lambda = 643$ nm in the case where $\varphi = 45°$ and for the quadratic fitting using the form $|t(k_r)| = ak_r^2$. The abbreviation "Sim." has a full name of "Simulation". **f** Simulated optical transfer function $|t(k_x)|$ (along the direction $k_y = -0.18k_0$, which is marked using the green line in (**d**)) for first-order differentiation and for linear fitting using the form $|t(k_x)| = b|k_x|$.

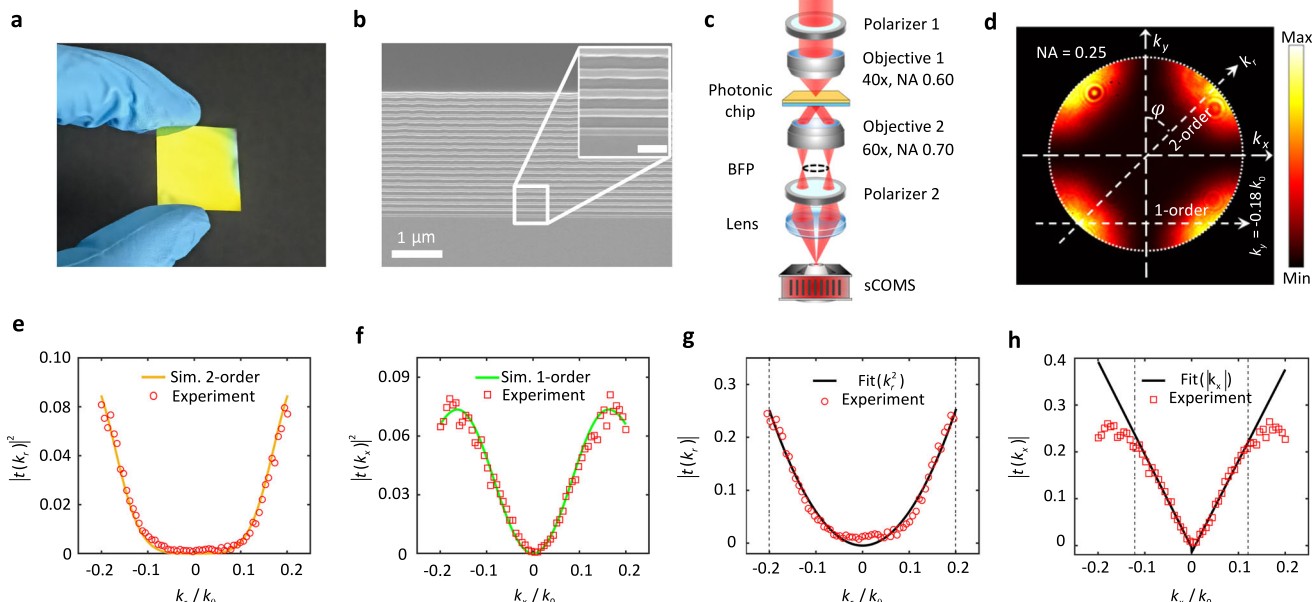

**Fig. 2 | BFP imaging of the planar photonic chip, no specimens are placed on the chip. a** Photograph of the fabricated planar photonic chip. **b** Cross-sectional SEM view of the photonic chip. **c** Schematic of the experimental setup used to perform BFP imaging. **d** Experimentally measured BFP image at $\lambda = 643$ nm. **e, f** Intensity profiles along the second-order and the first-order differentiation directions in (**d**), respectively, together with the corresponding simulated curves. **g, h** Optical transfer functions extracted from (**c**) and (**d**), respectively, using quadratic and linear fitting.

electric field in the central region (where $k_r \approx 0$) is nearly the same as that of the incident light. Then, if an analyzer with orientation perpendicular to the polarization direction of the polarizer is inserted between the photonic chip and the detector, the electric field intensity at the center (where $k_r = 0$) can be reduced to nearly zero and the directional transmission is removed (i.e., $t_0 = 0$), as verified by the yellow circular region shown in Fig. 1d. The OTF in this region can then be fitted using the function $t = ak_r^2$. It is anticipated that this type of nonlocality in the momentum domain (here, the nonlocality means that the optical transmission through the multilayer is dependent on the incident angle) enables the second-order spatial differentiation of the input field along the in-plane radial direction, as shown in Fig. 1e and in Fig. S1c,d in the supplementary information. The difference between $t_s$ and $t_p$ becomes larger in the region where $k_r$ is increasing, the polarization orientation of the transmitted electric field is obviously deflected compared to the incident polarization orientation. In this case, the analyzer not only removes the direct component, but also extends the range of quadratic line type to larger numerical apertures by modifying the transmission curve that enables the second-order differentiation.

It is worth noting that the polarization direction is not deflected along the horizontal (along $\mathbf{k_x}$) and vertical (along $\mathbf{k_y}$) directions, but is largely deflected along other azimuth angles. Therefore, the analyzer tangentially modulates the transmittance spectrum to get a transmission curve that satisfies the linear-line type, within the green trapezoidal regions (Fig. 1c). It means that here the OTF is of the form $t = b|k_x|$ or $t = b|k_y|$ when the optical field passing through this analyzer, as illustrated in Fig. 1f and Fig. S1e in the supplementary information. This indicates that the first-order spatial differentiation can be performed using the nonlocality of these regions. The fitting coefficients for both $a$ and $b$ are listed in Tables S1 and S2 in the supplementary information. By comparing the magnitudes of the coefficients $a$ and $b$, the position and size of the yellow circular and the green trapezoidal regions are determined (Fig. 1c).

The theoretical analysis on the origin of the multiple-order differentiations are described in details in the supplementary information (Sections 1 to 4). Here, this analysis is briefly described as follows. In

the condition that the polarization directions of the input and output field are perpendicular to each other ($\mathbf{e_{in}} = \begin{bmatrix} 0 \\ 1 \end{bmatrix}$ and $\mathbf{e_{out}} = \begin{bmatrix} 1 \\ 0 \end{bmatrix}$), the OTF has a form of (more details are shown in the Section 4 of the supplementary information):

$$t\left(k_x, k_y\right) = \frac{\sin(2\varphi)}{2}\left(t_p - t_s\right) \tag{3}$$

where the difference in transmission coefficients $t_s$ and $t_p$ (transmissivity for $s$- and $p$-polarized light through the chip) will induce the deflection of the output polarization direction from that of the incident polarization orientation (as shown in Fig. 1c). The term $\sin(2\varphi)$ related to azimuth angle $\varphi = \arctan(k_x/k_y)$ represents the angular modulation of transmissivity by the analyzer. Due to the horizontal symmetry of the planar structure, the Taylor expansion of the term $(t_p - t_s)$ can be written as $t_p - t_s = \sum_{n=1}^{N}(C_{p2n} - C_{s2n})\theta^{2n}$ where $C_{s2n(p2n)}$ are the coefficients of even terms of the Taylor expansion of $t_{s(p)}$ and $\theta = \arcsin(k_r/k_0) \approx k_r/k_0$. Considering the continuous variation of transmissivity with the frequency of the input light, the term $(t_p - t_s)$ is in the quadratic form and the other expansion terms are smaller at the wavelength of 643 nm, which satisfies $t_p - t_s = (C_{p2} - C_{s2})\theta^2$ (as shown in Figs. S2a, d). Thus, the OTF can be re-written as:

$$t\left(k_x, k_y\right) = \frac{\sin(2\varphi)}{2}\left(C_{p2} - C_{s2}\right)\theta^2 \tag{4}$$

Interestingly, when the azimuth angle $\varphi$ is fixed as a constant (along the direction of the diagonal dashed-line on Fig. 1d), the OTF has a form:

$$t = a \cdot k_r^2, \quad a = \frac{\left(C_{p2} - C_{s2}\right)}{2k_0^2} \cdot \sin 2\varphi \tag{5}$$

On the other hand, when the vertical wave vector $k_y$ is fixed as a constant (along the direction of the horizontal dashed-line on Fig. 1d),

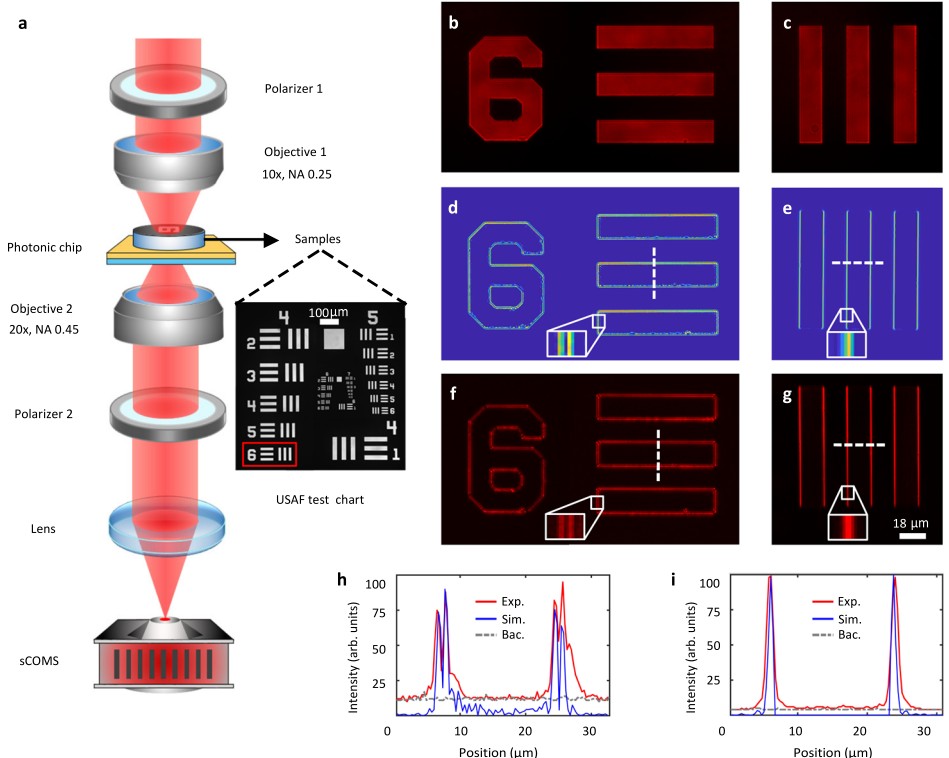

**Fig. 3 | Performance of the multiple-order spatial differentiation for amplitude objects. a** Schematic of the imaging set-up. The planar chip differentiator was placed directly below a standard 1951 USAF test chart (the sample, inset graph on the left corner). The target was illuminated using an out-of-focus linearly-polarized beam at a wavelength of 643 nm. **b**, **c** Imaging results for the target without the polarizer 2 (analyzer) being placed before the detector. **d**, **e** Calculated second-order and first-order derivatives of the images shown in (**b**) and (**c**), respectively. **f**, **g** Imaging results for the target when the polarizer 2 was placed before the sCMOS. The orientations of the two polarizers are perpendicular. The insets in (**d**), (**e**), (**f**), and (**g**) show magnified images of the regions marked using a line-box in each case. **h**, **i** Vertical and horizontal cuts through the images shown in parts (**d**)–(**g**) (white dashed lines), illustrating the consistency between the experimental and the calculated results. The grey dashed lines on (**h**) and (**i**) represent the background intensity. The full names of the abbreviations "arb. Units", "Exp." and "Bac." are "arbitrary units", "Experiment" and "Background", respectively.

the OTF will have another form derived from Eq. (4):

$$t = b \cdot k_x, \quad b = \frac{\left(C_{p2} - C_{s2}\right)}{k_0^2} \cdot k_y \quad (6)$$

Equation 5 indicates that the second-order differentiation can be performed along the in-plane radial direction (along $\mathbf{k_r}$), and Eq. 6 means that the first-order differentiation along the direction of $\mathbf{k_x}$ can be realized.

## Back focal plane imaging of the planar photonic chip

The above theoretical analysis and numerical calculations show that the same chip has two types of nonlocality, which means that both first-order and second-order spatial differentiation can be performed on the same photonic chip. To verify this point, the planar photonic chip shown in Fig. 2a was fabricated by plasma-enhanced chemical vapor deposition (PECVD). The fabrication process is described in greater detail in the section Methods, and a scanning electron microscope (SEM) image of the fabricated chip is shown in Fig. 2b. An in-house-fabricated back focal plane (BFP) imaging setup, which is shown in Fig. 2c, was used to characterize the angular-dependent transmission or nonlocality of this photonic chip[32].

The transmitted BFP image shown in Fig. 2d has a similar intensity distribution to that of the simulated image shown in Fig. 1d. For example, the extracted intensity distributions along the two dashed lines in Fig. 2d coincide with the two simulated curves shown in Fig. 2e, f. This consistency between the experimental and the

simulated BFP images ($\lambda$ = 643 nm) within a large $k$-space is illustrated in the images shown in Fig. S3a, b, e, f in the supplementary information. The experimental OTF that was extracted from these BFP images can be fitted using a parabolic curve (Fig. 2g, along the radial direction $\varphi$ = 45°), or fitted using a mirror-symmetrical line (Fig. 2h, along the direction $k_y = -0.18k_0$). The consistency between the experimental curves and the fitted curves clearly verifies that the chip has a nonlocality and thus will have the ability to perform both the first-order and second-order spatial differentiations.

## Spatial differentiation for imaging the amplitude objects

One of the primary benefits of the planar photonic chip is its ability to be vertically integrated with conventional optical systems. Here, the chip is incorporated into a standard transmission optical microscopy system. In this system, the chip is placed below the amplitude step objects of a standard 1951 USAF resolution test chart (inset graph on Fig. 3a). By using this chart with a step edge, there will be $n$ peaks at the edge of the image when $n$-order spatial differentiation is performed. Figure 3a shows a schematic of the imaging system, in which the sample is illuminated by an out-of-focus beam, and the different illuminated regions correspond to different angles of incidence. Imaging results acquired without the polarization analyzer for three elements on the test chart are shown in Fig. 3b, c, which can be regarded as bright field optical images of these elements. Figure 3d shows the calculated second-order derivative of these bright field images. Figure 3f shows images of the test chart acquired when the photonic chip was operating as a second-order differentiator and when the orthogonally polarized analyzer was inserted into the optical system, as

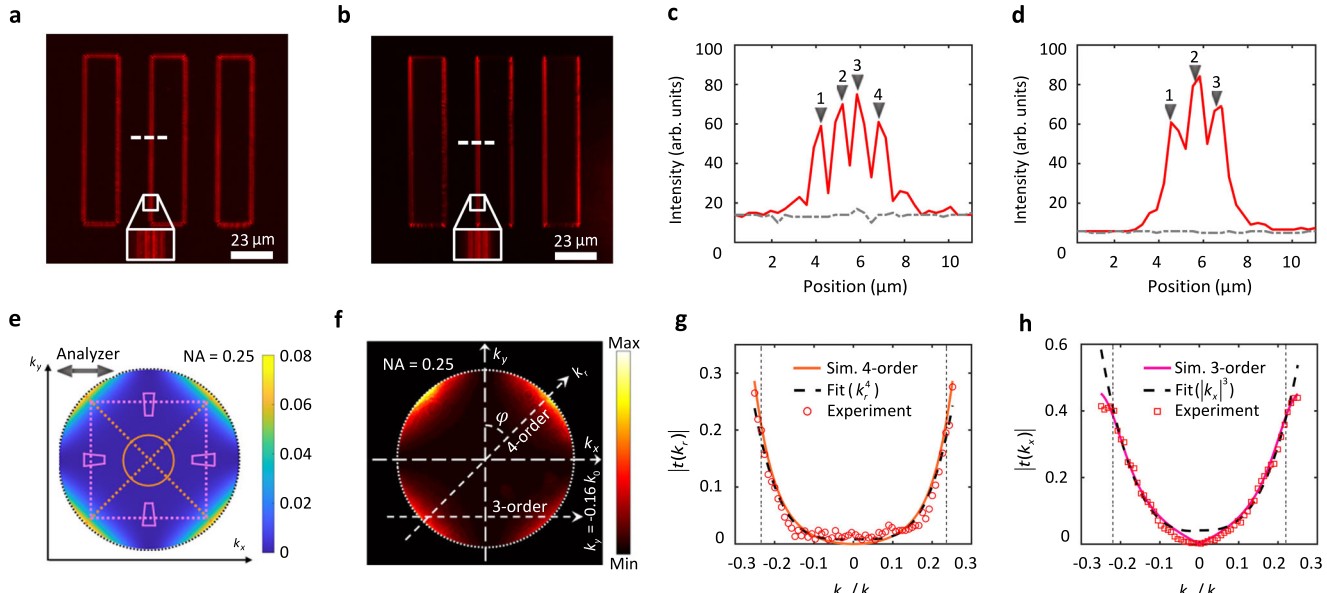

**Fig. 4 | The third- and fourth-order spatial differentiation at a wavelength of 638 nm. a, b** Imaging results for the target in the case of the fourth-order (**a**) and (**b**) the third-order spatial differentiation. The insets in (**a**) and (**b**) show magnified images of the regions marked using a line-box in each case. **c, d** Intensity profiles measured along the white dashed lines marked in (**a**) and (**b**). The grey dashed lines represent the background intensity. **e** Simulated BFP image at $\lambda = 638$ nm, the double arrow-headed line indicates the orientation of the analyzer. The trapezoidal regions are designed to perform the third-order differentiation, and the circular

region for the fourth-order differentiation. The vertical and horizontal pink dotted lines indicate the directions of the third-order differentiation, the tilt orange dotted line indicates the fourth-order differentiation. **f** Experimentally measured BFP image at $\lambda = 638$ nm. **g, h** Intensity profiles along the differentiation directions of the fourth-order (along $\varphi = 45°$) and the third-order(along $k_y = -0.16k_0$) in (**f**), together with the corresponding simulated curves, and quartic and cubic fitting curves, respectively.

demonstrated in Supplementary Video 1. The edges of the micrometer-scale elements are presented clearly along both the horizontal and vertical directions for both the experimental and simulated images, thus indicating the two-dimensional (2D) spatial differentiation process. Figure 3h shows the intensity profile along the dashed line through the differentiated image (shown in Fig. 3d, f), in which two closely spaced peaks are formed around each edge. This phenomenon represents the typical nature of the second-order derivative process[11,12].

In contrast, when the range of illumination angles for the incident beam (or for the incident wavevectors) is tuned to match the horizontal dashed line (where $k_y = -0.18k_0$ in $k$-space, in the momentum domain) as shown in Fig. 2d, the first-order differentiation of the image (both experimental and simulated) will occur as shown in Fig. 3e, g, i. One peak is formed around the edges, which represents the verification of the first-order derivative[13,25]. The edges of the elements are resolved clearly along the horizontal direction because the first-order derivative process is conducted along the $x$-axis. This is the one-dimensional (1D) first-order spatial differentiation process of the image. The image generation process is demonstrated intuitively in Supplementary Video 2. If the test chart is rotated around the optical axis of the imaging objective by 45°, then the 2D first-order spatial differentiation can be performed on the image as shown in Fig. S4 in the supplementary information. In this case, the edges of the elements are resolved clearly along both the horizontal and vertical directions, with only one peak formed around the edges.

To quantify the resolution of the planar photonic chip for edge-enhancement imaging experimentally, a series of micrometer-scale elements on the resolution test chart were imaged as shown in Fig. S5a-g. The edges of elements with dimensions as small as 2 μm were resolved (Fig. S5g, h), thus indicating the possibility of 2D spatial differentiation with resolution of less than 2 μm. Detailed analysis on the resolution of the planar chip-based differential system is given in "Section 5" and Fig. S6 of the supplementary information.

## The higher-order spatial differentiations by the same chip

One interesting phenomenon is observed when the working wavelength is tuned to 638 nm, the coefficients' difference in transmission for $s$- and $p$-polarized light satisfies $t_p - t_s = (C_{p4} - C_{s4})\theta^4$ (as shown in Fig. S2b, e), so that a form of the OTF can be obtained as follows:

$$t\left(k_x, k_y\right) = \left(C_{p4} - C_{s4}\right)\frac{\sin(2\varphi)}{2}\theta^4 = \left(C_{p4} - C_{s4}\right) \cdot \left[\left(\frac{k_x}{k_0}\right)^3\frac{k_y}{k_0} + \frac{k_x}{k_0}\left(\frac{k_y}{k_0}\right)^3\right]$$

(7)

When the azimuth $\varphi$ is fixed as a constant, the OTF has a form of:

$$t = c \cdot k_r^4, \quad c = \frac{\left(C_{p4} - C_{s4}\right)}{2k_0^4} \cdot \sin2\varphi$$

(8)

Then, the two-dimensional fourth-order spatial differentiation can be achieved, as illustrated in Fig. 4a and shown in Supplementary Video 3. Figure 4c shows the intensity profile along the dashed line through the differentiated image (shown in Fig. 4a), in which four closely spaced peaks are formed around the edge. This phenomenon represents the typical nature of the fourth-order derivative process. Similarly, when the $k_y$ is a constant (it can be experimentally realized by tuning the illumination directions), another form of the OTF is revealed:

$$t = d \cdot k_x^3, \quad d = \frac{\left(C_{p4} - C_{s4}\right)}{k_0^4} \cdot k_y$$

(9)

Then, the third-order spatial differential image can be obtained as shown in Fig. 4b and Supplementary Video 4. The intensity profile (Fig. 4d) along the dashed line through the differentiated image

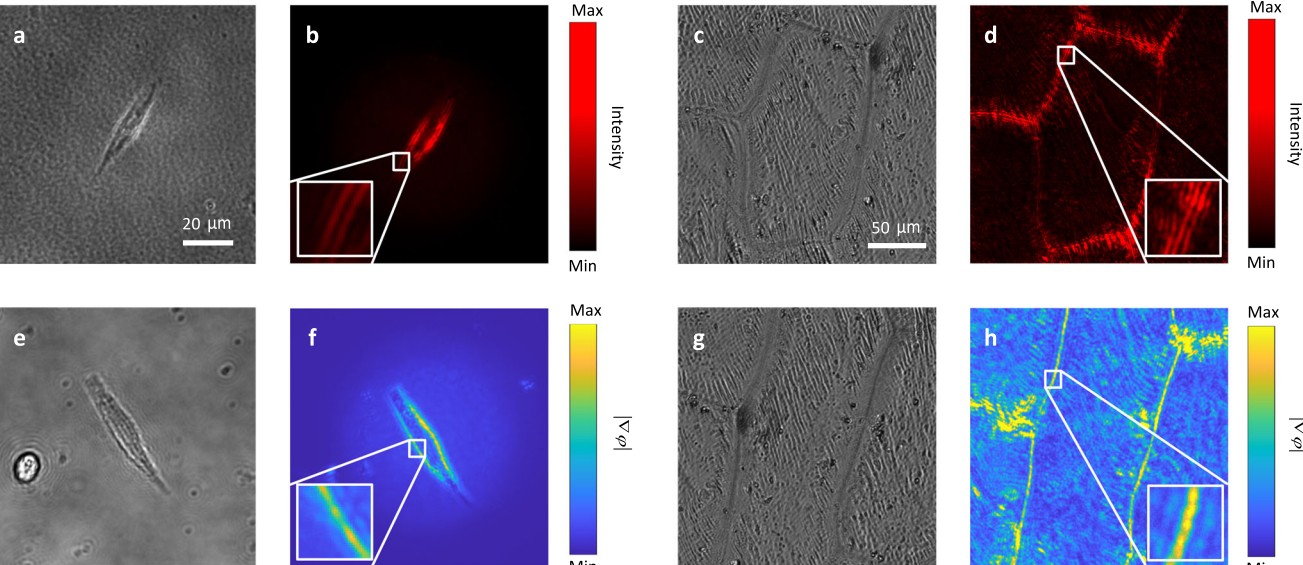

**Fig. 5 | Performance of multiple-order spatial differentiation for phase objects.** Bright field optical images of (**a**), (**e**) B16F10 melanoma cells and (**c**), (**g**) onion epidermis. **b**, **d** The second-order spatially differentiated images of the B16F10 melanoma cells and the onion epidermis, respectively. **f**, **h** The first-order spatially differentiated images of B16F10 melanoma cells and onion epidermis, respectively. The insets in (**b**), (**d**), (**f**), and (**h**) show magnified images of the regions marked using a line-box in each case.

(Fig. 4b), shows that three closely spaced peaks are formed around the edge, which is the typical nature of the third-order derivative process.

The simulated and experimental BFP images at the wavelength of 638 nm are presented in Fig. 4e f, which are consistent with each other. This consistency between the experimental and simulated BFP images ($\lambda$ =638 nm) within a large $k$-space is illustrated in the images shown in Fig. S3c, d, g, h in the supplementary information. On the simulated BFP image (Fig. 4e), the pink trapezoidal regions and orange circular region are designed for the third-order and fourth-order spatial differentiation, respectively, which are of the same principle as that for the first- and second-order differentiation, as demonstrated in Fig. S1f, g, h. As shown in Fig. 4g, h, the simulated OTF for the fourth-order differentiation (at $\varphi = 45°$) and the third-order differentiation (at $k_y = -0.16k_0$) can be matched with the fitting curves in the form of $t = ck_r^4$ and $t = d|k_x|^3$, respectively. They are also consistent with the corresponding experimental curves derived from the experimental BFP image.

By using the same principle to tailor the angular transmission of this planar chip at other wavelengths, the fifth- and sixth-order differential operations can be implemented as shown in Fig. S3c, f in the supplementary information. To the best of our knowledge, third- and fourth-order spatial differentiation have never been experimentally realized before. Here, they are successfully demonstrated with one optical element. Based on the proposed theoretical analysis, the higher order spatial differentiation can also be realized with this chip. Spatial differentiation is one of the basic mathematical operation, and realization of different orders derivative with single optical element will be of great importance for optical analog signal processing. More importantly, combining multi-order differentiation operations, the Taylor expansion can be implemented on the electromagnetic field. In other words, optical analogue computing can be used to realize the Taylor expansion operation on the wave function of the input optical field.

In addition, the working wavelength for each-order spatial differential operation can be tuned by varying the thickness of each layer in the planar chip (More details are given in Section 6 of the supplementary information). For examples, 7 planar chips were fabricated via PECVD, whose operating wavelengths for second-order differential operation can be tuned precisely from 619 nm to 627 nm, as shown in

Fig. S7 in the supplementary information. At these operating wavelengths, the edges of the microscale elements can be resolved clearly. Similar as those shown in Fig. 4, based on the theoretical analysis described in the supplementary information, higher-order spatial differentiation can also be realized with these 7 planar chips, which will work in the desired wavelengths.

## Spatial differentiation for imaging the phase objects

In this section, biological samples (B16F10 melanoma cells and onion epidermis) were used as imaging specimens. These samples only have a significant effect on the phase and not on its amplitude, and this class of specimens is thus referred to as a phase object class[9,16,33]. The experimental setup used is the same as that shown in Fig. 3a, with the exception that the samples on the planar photonic chip were replaced with the biological specimens. Figure 5a, e show the bright field images of the melanoma cells, and Fig. 5c, g show the bright field images of the onion epidermis. These bright field images were obtained using a broad-band white light illumination source. The shapes and boundaries of the two specimens are not very clearly discernible in the bright field images because of their transparent nature. When the illumination wavelength is tuned to 643 nm, the second-order (Fig. 5b, d) and first-order (Fig. 5f, h) differentiated images can then be obtained, and these images show both significant edge enhancement and high-contrast cell boundaries.

Dark-field microscopy can also enhance the edges of these specimens, but it requires the use of complex components such as a condenser annulus, and also places high requirements on alignment of the optical path. Here, use of the planar photonic chip can reduce the system complexity significantly. The chip acts as a coverslip that provides edge enhancement automatically without use of additional optics. However, for conventional dark-field microscopy, the NA of the objective can be up to 0.70 in routine. For all the edge-enhancement imaging techniques based on the spatial differentiation, the allowed NA of the imaging objective (allowed angle-range for spatial differentiation) is smaller, such as NA less than 0.40 or even smaller[8,9,12,25]. It means that the conventional dark-field microscopy can achieve better spatial resolution. On the other hand, it also means that a high NA (larger than 0.70) condenser for dark-field illumination is required for the conventional dark-field microscope, which will reduce the

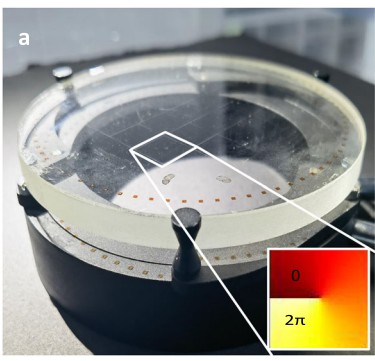
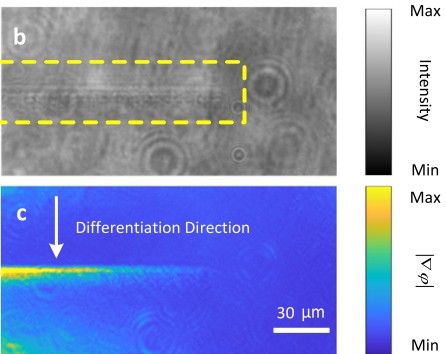
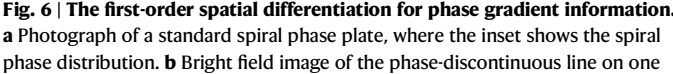

**Fig. 6 | The first-order spatial differentiation for phase gradient information.** **a** Photograph of a standard spiral phase plate, where the inset shows the spiral phase distribution. **b** Bright field image of the phase-discontinuous line on one element of this plate. **c** The first-order spatially differentiated image of this phase-discontinuous line.

illumination region and also the field of view of the microscope. This will not happen for the planar chip based dark-field imaging, because that the use of lower NA imaging objective will bring larger field of view and also there is no need for a larger NA condenser to realize the dark-field illumination.

Here, when the first-order spatial differentiation is performed on the phase object (e.g., the transparent biological cell), the phase information of the specimen is obtained as follows[25]. The wave field that passes through the phase object can be written as $E_{in} = 1 \times e^{i\phi(x,y)}$ (the input intensity $I_{in} = E_{in} \times E_{in}^* = 1$, showing no contrast). After the first-order differentiation operation (i.e., transmission through the planar chip), the output electric field can be written as $E_{out} = \partial E_{in}/\partial x = ie^{i\phi(x,y)} \left[\partial\phi(x,y)/\partial x\right]$, and the corresponding intensity $I = |\partial\phi(x,y)/\partial x|^2$, which shows phase contrast that is proportional to phase gradients in the specimen. Therefore, the first-order differentiated image contains the phase information for this specimen.

For example, Fig. 6a shows a commercial phase-type diffractive optical element called a vortex phase plate (topological charge =1, VPP-1b, RPC Photonics, Inc., USA) with an optical thickness that is proportional to the azimuth angle rotation. This element is always used to generate optical vortex beams[34,35]. There is a line that indicates the discontinuous spiral phase (i.e., the boundary between the spiral phase of 0 and the spiral phase of $2\pi$; inset of Fig. 6a) on this plate. This line is very blurred on the bright field image (Fig. 6b), while in contrast, the line becomes obvious in the first-order differentiated image (Fig. 6c) because of the existence of the phase gradient across this line.

It is known that most of the important features of specimens can be preserved even if only the phase is retained[36], and thus retrieval of the optical phase information is highly important in many fields, particularly in biological science. For commonly used phase-contrast techniques such as differential interference contrast (DIC) and phase contrast microscope, of which the configuration is much more complex and precise alignment of the optical path is necessary. On the contrary, the proposed planar chip is both compact and easy to use. It does not mean that the proposed method should replace the conventional dark-field or phase contrast imaging techniques in all areas. The obvious advantages of the chip-based imaging technique are low cost, easy to use and large field of view. The same chip can be used as a cover slip and easily incorporating to the widely used optical microscopy to realize the functions of both the dark-field imaging and the phase information imaging.

## Discussion

In summary, a planar photonic chip composed of a well-designed dielectric multilayer structure is proposed to experimentally realize multiple mathematical operations, i.e., the first-, second-, third-, and fourth-order spatial differentiations, without the need to change the chip's structural parameters. The principle lies in the angular-dependent responses or nonlocality encoded within this dielectric multilayer structure. Through appropriate engineering of the non-locality of this multilayer structure, the angle-dependent transmission along one direction in the momentum domain can match the requirements for one mathematical operation, and the other directions can be used to perform other operations. A theoretical analysis on the origin of these multiple spatial differentiations with single chip is proposed, which are consistent with the experimental results. It also reveals that higher order spatial differentiations can be achieved with the same chip. We have proved that multi-order differential operations can be imparted on the optical waves with one optical element so that this single multilayer chip can operate as a multifunctional compact device for optical analog signal processing. More importantly, the proposed approach also can realize the Taylor expansion of a mathematic function, which has never been proposed or realized before. Benefit from the ability for multiple-orders spatial differential operations, it is possible to use the planar photonic chip for solving the differential equations with constant coefficients[37–39] and conducting beam shaping[15,40,41]. In addition, higher-order differentiations can lead to sharper peaks at the edges of the image, meaning that higher-order differentiation can play an important role in image sharpening[24].

Because of its ability to perform spatial differentiation and the thin film factor, this planar chip can be incorporated easily into conventional imaging systems, thus enabling high-contrast edge imaging of both amplitude and phase objects. For pure phase objects (e.g., biological cells), the phase information in the spatial domain can also be derived. This dielectric multilayer structure does not require precise nanofabrication procedures and can be manufactured on a large scale at low cost. The chip working as spatial differentiator offers the advantage of a multifunctional wave-based analogue computing ability, thus it will provide a route for designing fast, power-efficient, compact and low-cost devices used in edge detection and optical image processing, and offers opportunities for the rapid developing research field where the engineered substrates are proposed to enhance the imaging performance of the conventional optical microscope[12,42–48].

## Methods

### Method of fabricating the photonic chip

The photonic chip was fabricated via PECVD (Oxford System 100, UK) of $SiO_2$ and $Si_3N_4$ layers on a standard microscope cover slip (thickness: 0.17 mm) at a vacuum of <0.1 mTorr and a temperature of 300 °C. Before the coating process, the coverslip was cleaned using acetone, absolute ethanol, piranha solution, and nanopure deionized water in turn, and was then dried using a $N_2$ stream. The $SiO_2$ layer is the low refractive index dielectric layer and the $Si_3N_4$ layer is the high refractive

index dielectric layer. The process of $Si_3N_4$ generation is dependent on the chemical reaction of $SiH_4$ with $N_2O$ and $NH_3$ at high temperatures, and the process of $SiO_2$ generation is dependent on the chemical reaction of $SiH_4$ and $N_2O$. By controlling the ventilation volumes and ventilation rates of the various gases, the thickness and refractive index of each layer can be determined precisely. To confirm that the designed optical functions of the fabricated photonic chip were realized, an in-house-built reflection BFP imaging setup was used to characterize the photonic band gaps of this dielectric multilayer structure (see Fig. S8).

## Preparation of biological samples

The live biological cell is B16-F10, a mouse melanoma highly transfected cell which is a subline of B16-F0 cell and the cells for experiments were cultured in DMEM with 10% FBS and 1% PS. The onion cells were peeled from the upper epidermis and immersed in physiological saline solution.

## Optical experiments

All optical measurements were performed using a modified optical microscope (Ti2-U, Nikon, Japan). In the transmission BFP imaging setup (Fig. 2c), the illumination beam with a central wavelength of 643 nm and a bandwidth of 2 nm was emitted from a supercontinuum fiber laser with an acousto-optic tunable filter (SuperK EXU-6, NKT Photonics, Denmark). The beam was collimated and passed through polarizer 1. Then, the beam was focused onto the photonic chip using objective 1 (S Plan Fluor, 40×, numerical aperture NA = 0.60; Nikon, Japan). The transmitted signals were collected using objective 2 (S Plan Fluor, 60×, NA = 0.70; Nikon, Japan). The analyzer (polarizer 2), which has an orientation that is perpendicular to that of polarizer 1, was placed between objective 2 and the tube lens. It was then used to eliminate the direct component of the transmitted signal and modify the transmission curves. The BFP images were then recorded using a Neo scientific complementary metal-oxide-semiconductor (sCMOS) detector (Andor Oxford Instruments, UK).

In the front focal plane imaging set-up (Fig. 3a), the illumination beam remained unchanged. It was defocused using an objective (UPlanSApo, 10×, NA = 0.25; Olympus, Japan) onto the specimens that were placed on the photonic chip. This defocused beam can provide a variety of illumination angles in different areas, and can thus induce different mathematical operations as a result. The scattered and transmitted light from the specimens and the photonic chip was collected using objective 2 (S Plan Fluor, 20×, NA = 0.45; Nikon, Japan), and was then imaged onto the other camera (Andor Oxford Instruments, United Kingdom) with the aid of a tube lens. An analyzer (polarizer 2) was inserted between the tube lens and objective 2.

## Data availability

The data that support the plots within this paper and other finding of this study are available from the corresponding author upon request. Source data for Figs. 1–6 are available at https://doi.org/10.6084/m9.figshare.21506229.v3.

## Code availability

The codes that support the findings of this study are available from the corresponding authors upon reasonable request.

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

## Acknowledgements
This work was supported by the National Key Research and Development Program of China (2021YFA1400700), the National Nature Science Foundation of China (grant nos. 12134013 and 62127818), the Hefei Municipal Natural Science Foundation (grant no. 2021007), the Key Research & Development Program of Anhui Province (202104a05020010), and the Fundamental Research Funds for the Central Universities (WK2340000109), Inovation Program for Quantum Science and Technology (No. 2021ZD0303301). D.G. Zhang is supported by a USTC Tang Scholarship. The work was partially performed at the University of Science and Technology of China's Center for Micro and Nanoscale Research and Fabrication.

## Author contributions
D.G.Z. and Y.L. initiated the work. Y.L., M.C.H., and Q.K.C. acquired the theoretical and simulated results. Y.L. fabricated the samples. Y.L. and D.G.Z. performed the optical experiments. D.G.Z. and Y.L. wrote the manuscript. D.G.Z. supervised the work. All authors discussed the results and commented on the manuscript.

## Competing interests
The authors declare no competing interests.

## Additional information

**Peer review information** : *Nature Communications* thanks Amin Khavasi, Hailu Luo and the other, anonymous, reviewer(s) for their contribution to the peer review of this work. Peer reviewer reports are available.

