## [Peer Review File · Nature Communications]

Single planar photonic chip with tailored angular transmission for multiple-order analog spatial differentiatorREVIEWER COMMENTS

Reviewer #1 (Remarks to the Author):

This work presents a novel approach for implementing first and second order differentiation with a single chip. The proposed chip is a multilayer structure with no need to patterning and thus lithography. The authors have demonstrated the performance of the device experimentally. I think this work can be published in Nature Comm. due its significant contribution to the field of analog optical computing. There are two minor comments for further improvement of the work:

1- It would be useful if the authors obtained the FOM of their structure based on the approach introduced in:

Karimi, Parisa, et. al., Optics Express 28, no. 2 (2020): 898-911.

2- There are some related works which are not cited (I recommend citing some of these works):

A) Xu D, He S, et. al. Optics Letters. 2020 Dec 15;45(24):6867-70.

B) Zhou C, et. al. Optics Express. 2019 Sep 16;27(19):27295-307.

C) Kwon H, et. al. ACS Photonics. 2020 Jun 9;7(7):1799-805.

D) Mohammadi H, et. al. Optics Express. 2022 May 23;30(11):17806-23.

Reviewer #2 (Remarks to the Author):

During the past few years, optical metamaterials and metasurfaces have been suggested to perform analog spatial differentiation for edge detection which show superior integration capability compared with the traditional bulky system comprising lenses and spatial filters. A suitably designed metamaterial structure was theoretically proposed to perform desired mathematical operations including edge detection as light propagates through it. Deliberately designed layered structure was also suggested for spatial differentiation. In this manuscript, a dielectric multilayer structure is proposed to operate as both first- and second-order spatial differentiator without need to change the structural parameters. This planar nanostructure offers the advantages of a thin form factor and a multifunctional wave based analogue computing ability, which will bring opportunities in optical imaging and computing. Therefore, I think this manuscript fit for the publication in Nature Communications. However, there are several issues in the paper need to be elucidated further before the acceptance for publication.

1. It is known that an integrated circuit or monolithic integrated circuit (also referred to as an IC, a chip, or a microchip) is a set of electronic circuits on one small flat piece (or "chip") of semiconductor material, usually silicon. Large numbers of tiny MOSFETs (metal oxide semiconductor field effect transistors) integrate into a small chip. This results in circuits that are orders of magnitude smaller, faster, and less expensive than those constructed of discrete electronic components. However, the photonic chip in this manuscript is constructed from a layered nanostructure. I think the multilayer structure proposed in the present manuscript is just like a one-dimensional photonic crystal filter or Bragg grating. Different resonance would be achieved for different incident conditions. Therefore, the concept of photonic chip seems to me is overhyped.

2. I would like to draw the author's attention to the work on analog spatial differentiation with phase-shifted Bragg grating. For example

(1) Spatial differentiation of optical beams using phase-shifted Bragg grating
Optics Express 22, 025084 (2014).

(2) Optical computation on of the Laplace operator using phase-shifted Bragg grating
Optics Letters 39, 1278 (2014).

In the two papers, the first-order and second-order differentiation have been realized by layered nanostructures. The proposed approach can be used to obtain other types of beam spatial transformations. This can be achieved with the use of resonant diffraction gratings having more complex form of the reflection or transmission coefficient. A more quantitative description the theoretical model or some comparative discussion will help the reader more easily understand the

superiority of the proposed scheme.

3. It is not clear how to switch between different order differential operations. Is it possible to have both 1st and 2nd order differentiation in the output field and how to distinguish them? Why are the wavelengths of first and third order differential operations different? The authors claim that “Third- and fourth-order differentiations that have never been realized before.” I suggest the authors add some discussions to clarify why it is important to realize the higher-order differentiations.

In addition, the authors should check the manuscript carefully. For example, figure 3b is not clearly depicting the areas of the edges, the authors should provide high resolution image. The scale bar on Figure 6 is missing.

Reviewer #3 (Remarks to the Author):

The authors propose multiple-order analog spatial differentiator with simple multilayer structures. By designing the spectrum transfer function of a single device, different order spatial differentiations are realized at different incident angles and different wavelength. I think that it is quite interesting that a simple structure can be used to perform the optical computation of multiple-order differentiation. As the experimental demonstrations, such a device is also useful for the image processing and surely other applications. Below there are some of my comments for the authors' consideration.

- What resolution for such high-order differentiation, the three- and four-order differentiation? Any advantages over the first- or the second-order differentiation?

- Since the different polarizations of the incident and the output light are analyzed to realize the differentiation, does the spin Hall effect of light also contribute to the spatial differentiation, such as Physical Review Applied 11, 034043 (2019) and Advanced Photonics 2, 016001 (2020)?

- Any discussion or general suggestion to design the structure for the other operation wavelength?

A point-by-point response to the referees' comments

REVIEWER COMMENTS

Reviewer #1 (Remarks to the Author):

This work presents a novel approach for implementing first and second order differentiation with a single chip. The proposed chip is a multilayer structure with no need to patterning and thus lithography. The authors have demonstrated the performance of the device experimentally. I think this work can be published in Nature Comm. due its significant contribution to the field of analog optical computing.

Response: Thank you very much for the positive comments.

There are two minor comments for further improvement of the work:

1- It would be useful if the authors obtained the FOM of their structure based on the approach introduced in:

Karimi, Parisa, et. al., Optics Express 28, no. 2 (2020): 898-911.

Response: Thank you for your valuable suggestion. The suggested literature improves our understandings on the optical resolution of multi-order spatial differentiation. We cited this article and added a section on the resolution of multi-order differentiation in the supporting information to improve the quality of our work.

For your convenience, we copy these contents here.

“Section 5: Resolution of multi-order differentiation

An issue worthy of attention is “What is the resolution of such multi-order differentiations?”. Inspired by the description of figure of merit (FOM) in the imaging process, we proposed a method to define the resolution of multi-order differentiations.

A one-dimensional rectangular pulse wave function $E_{in}^x(x) = p(x) = \Pi(x/2x_0)$ (Fig. S6a) is used to replace the field with information of USAF Resolution Test Target, which also can be regarded as the sum of a large number of plane waves:

$$E_{in}^x(x) = \int_{-\infty}^{\infty} P(k_x) e^{-ik_x x} dk_x \quad (\text{S5.1})$$

where $P(k_x)$ is the spatial spectra of the incident field obtained using the spatial Fourier transform:

$$\begin{aligned}
P(k_x) &= \frac{1}{2\pi} \int_{-\infty}^{\infty} p(x) e^{ik_x x} dx \\
&= \frac{\sin k_x x_0}{\pi k_x} \\
&= \frac{x_0}{\pi} \text{sinc}\left(\frac{x_0}{\pi} k_x\right)
\end{aligned} \tag{S5.2}$$

The output field E_{out}^x modulated by OTF is calculated as:

$$E_{out}^x(x) = \int_{-k_{max}}^{k_{max}} t(k_x) P(k_x) e^{-ik_x x} dk_x \tag{S5.3}$$

where the OTF is written as $t(k_x) = ik_x^n$ where n represents the order of multi-order differentiations and k_{max} is the maximum transverse wave vector acceptable by the system, which is determined by the aperture of the collection system.

The normalized output fields $|E_{out}^x|^2$ around the right edge of the rectangle input wave are plotted in **Fig. S6b-S6e** where the number of edges corresponding to the order of the multi-order differential and the diffraction fringes are displayed. The definition of resolution can refer to the Rayleigh criterion in which the edges are only detectable until the maximum point of one edge signal coincides with the first zero of another edge signals. Therefore, we define the distance between the peak of the peripheral edge and the position of the first zero value as Δx_n , the distance between center and boundary of the edge is L_n , and the resolution of multi-order differentiation is obtained as $R = (2L_n - \Delta x_n) / \lambda_0$ (normalized to the incident wavelength λ_0). Finally, the theoretical resolution of the first-, second-, third- and forth- order differential are 1.10 (0.71 μm), 2.35 (1.51 μm), 3.52 (2.25 μm) and 4.64 (3.00 μm), respectively. Moreover, the analysis results of resolution in **Fig. S6b-S6e** are consistent with that in **Fig. 3i** and **3h** and in **Fig. 4d** and **4c**.

Figure S6. Analysis on the resolution of multi-order differentiation based on Rayleigh criterion. (a) Normalized rectangle incident field $|E_{in}^x|$ with “Left Edge” and “Right Edge” marked. (b)-(e) Normalized output field $|E_{out}^x|^2$ modulated by (b)first-, (c)second-, (d)third- and (e)forth-order differentiation around the “Right Edge” of the input field. Δx_n represents the distance between the peak of the peripheral edge and the position of the first zero value, and L_n represents the distance between center and boundary of the edge, where n is the order of the multi-order differentiations.”

2- There are some related works which are not cited (I recommend citing some of these works):

A) Xu D, He S, et. al. Optics Letters. 2020 Dec 15;45(24):6867-70.

B) Zhou C, et. al. Optics Express. 2019 Sep 16;27(19):27295-307.

C) Kwon H, et. al. ACS Photonics. 2020 Jun 9;7(7):1799-805.

D) Mohammadi H, et. al. Optics Express. 2022 May 23;30(11):17806-23.

Response: Thank you very much for your kind reminder. We are sorry that we have not noted these related works. All these papers are cited in the revised manuscript. Thank you again.

Reviewer #2 (Remarks to the Author):

During the past few years, optical metamaterials and metasurfaces have been suggested to perform analog spatial differentiation for edge detection which show superior integration capability compared with the traditional bulky system comprising lenses and spatial filters. A suitably designed metamaterial structure was theoretically proposed to perform desired mathematical operations including edge detection as light propagates through it. Deliberately designed layered structure was also suggested for spatial differentiation. In this manuscript, a dielectric multilayer structure is proposed to operate as both first- and second-order spatial differentiator without need to change the structural parameters. This planar nanostructure offers the advantages of a thin form factor and a multifunctional wave based analogue computing ability, which will bring opportunities in optical imaging and computing. Therefore, I think this manuscript fit for the publication in Nature Communications.

Response: We really appreciate these positive comments.

However, there are several issues in the paper need to be elucidated further before the acceptance for publication.

1. It is known that an integrated circuit or monolithic integrated circuit (also referred to as an IC, a chip, or a microchip) is a set of electronic circuits on one small flat piece (or "chip") of semiconductor material, usually silicon. Large numbers of tiny MOSFETs (metal oxide semiconductor field effect transistors) integrate into a small chip. This results in circuits that are orders of magnitude smaller, faster, and less expensive than those constructed of discrete electronic components. However, the photonic chip in this manuscript is constructed from a layered nanostructure. I think the multilayer structure proposed the present manuscript is just like a one-dimensional photonic crystal filter or Bragg grating. Different resonance would be achieved for different incident conditions. Therefore, the concept of photonic chip seems to me is overhyped.

Response: Thank you for your comment and reminder.

The chip of electronics is an integrated circuit on a wafer, which is composed of a large number of MOSFETs. By combining MOSFETs cells, differentiators and integrators are formed on the chip to perform various operations on the input binary data.

The one-dimensional photonic crystal (or dielectric multilayer) used in this work takes a pair of high and low refractive index layers as a cell, which is periodically stacked on a cover glass, and has the characteristics of miniaturization and integration. As you said, for different incident states, it has different response characteristics, so as to produce the effect of a multi-order differentiator. In the above aspects, the proposed structure is similar to the chip in electronics, so it is called "chip" in the previous manuscript.

Our one-dimensional photonic crystal is an ultra-thin plate-shaped structure coated with multilayer films on the cover glass, and by artificially designing the photonic band structure, this plate-shaped structure can realize more than the differential operations in the future.

At the same time, in our opinions, this name "photonic chip" can attract the more attention from a wider audience

of the journal Nature Communications. Therefore, we think it is appropriate to call this structure as a “chip”.

To our knowledge, these articles below also use the term “photonic chip” to describe the similar planar structures as ours. In these literatures, planar films containing gratings, waveguides or other film structures were used to realize some interesting experiments.

- (1) Chip-based wide field-of-view nanoscopy, *Nature Photonics* **11**, 322–328 (2017)
- (2) Structured illumination microscopy using a photonic chip, *Nature Photonics* **14**, 431–438 (2020)
- (3) Luminescent surfaces with tailored angular emission for compact dark-field imaging devices, *Nature Photonics* **14**, 310–315 (2020)
- (4) Dark field on a chip, *Nature Photonics* **14**, 266–267 (2020)

But if you insist on calling this structure as “filter” or “Bragg grating”, we are willing to make this change.

Thank you again for your valuable comments.

2. I would like to draw the author's attention to the work on analog spatial differentiation with phase-shifted Bragg grating. For example

- (1) Spatial differentiation of optical beams using phase-shifted Bragg grating, *Optics Express* **22**, 025084 (2014).
- (2) Optical computation on of the Laplace operator using phase-shifted Bragg grating, *Optics Letters* **39**, 1278 (2014).

In the two papers, the first-order and second-order differentiation have been realized by layered nanostructures. The proposed approach can be used to obtain other types of beam spatial transformations. This can be achieved with the use of resonant diffraction gratings having more complex form of the reflection or transmission coefficient. A more quantitative description the theoretical model or some comparative discussion will help the reader more easily understand the superiority of the proposed scheme.

Response: Thank you for your valuable suggestion.

These literatures are very interesting and improve our understanding on this topic. All the literatures are cited in the revised manuscript.

In these interesting literatures, the fundamental mode Gaussian beam is transformed into Hermite-Gaussian beam or Laguerre-Gaussian beam by various differential operations. This can be understood, because the fundamental mode Gaussian beam is the solution of the Helmholtz equation, and the combination of its spatial derivation results is also the solution of the differential equation, so Hermite-Gaussian beam E_{nm}^H and Laguerre-Gaussian beam

E_{nm}^L can be generated from the fundamental mode Gaussian beam E_0 : $E_{nm}^H = \omega_0^{n+m} \frac{\partial^n}{\partial x^n} \frac{\partial^m}{\partial y^m} E_0$ and

$$E_{nm}^L = k^n \omega_0^{2n+m} e^{ikz} \frac{\partial^n}{\partial z^n} \left(\frac{\partial}{\partial x} + i \frac{\partial}{\partial y} \right)^m (E_0 e^{-ikz}).$$

The well-designed one-dimensional photonic crystal proposed in our work can realize the multiple differential operations with only one photonic chip, which have not been reported in these two literatures. The other difference is that the structure used in our experiment is working in the transmission-mode, but not the reflection mode, which is more suitable for cascading multiple chips. By cascading, the combination of x-direction differential and y-direction differential of various order can be realized. As a result, more diversified Hermite-Gaussian beams can be realized.

3. It is not clear how to switch between different order differential operations. Is it possible to have both 1st and 2nd order differentiation in the output field and how to distinguish them? Why are the wavelengths of first and third order differential operations different? The authors claim that “Third- and fourth-order differentiations that have never been realized before.” I suggest the authors add some discussions to clarify why it is important to realize the higher-order differentiations.

Response: Thank you for raising these insight comments. A point to point response to these comments are given below.

(1) It is not clear how to switch between different order differential operations.

Response: This question can be answered in combination with Figure 1d and Figure 4e in the main text (For your convenience, we copy the figures below) that are back focal plane images, and the positions on the images represent

the radial $\theta = \arcsin\left(\frac{|k_r|}{|k_0|}\right)$ and azimuth $\varphi = \arctan\left(\frac{k_x}{k_y}\right)$ angles of the incident light. Then, the box and

circular area represent the angle of incidence. The incident light in this work is a focused light, but the focal point is not on the sample plane. In this way, different positions of the illumination spot on the sample represent different incident angles. In order to switch between differential effects of different orders, we need to move the sample to the corresponding illumination area and select the appropriate wavelength.

(2) Is it possible to have both 1st and 2nd order differentiation in the output field and how to distinguish them?

Response: This idea can be implemented. If we replace one focused beam used in our work with two incoherent beams, and conduct oblique and normal incidence respectively at the same position on the sample (the angle corresponding to the square box and circular area in Figure 1d), we can carry out first-order and second-order differentiation for the sample information at the same time and same position. In the scheme above, both 1st- and 2nd-order differentiation are implemented at the same position, so that the modulated fields will be added together to generate a new field. This new field has new meaning, such as the solution of differential equation. I think it is more useful to use them together than separate them. In the scheme proposed in our work, the 1st- and 2nd-order differentiation can occur simultaneously, and the differential domains are separated in space. Thus, we can distinguish them by the number of edges of amplitude type sample.

(3) Why are the wavelengths of first and third order differential operations different?

Response: The order of the differential depends on the optical transfer function (OTF), which has the form of $t(k_x, k_y) = \frac{\sin 2\varphi}{2}(t_p - t_s)$. What can be obtained from this formula is that the difference in transmission

coefficients t_p and t_s (transmissivity for p- and s-polarized light through the chip) determines the form of OTF.

When the difference $t_p - t_s = (C_{p2} - C_{s2})\theta^2$, we have $t(k_x, k_y) = \frac{\sin 2\varphi}{2}(C_{p2} - C_{s2})\theta^2$ where C_{pn} and

C_{sn} are the Taylor expansion coefficients of t_p and t_s , and then second-order differential and first-order differential derived by second-order differential can be realized. Similarly, when the difference

$t_p - t_s = (C_{p4} - C_{s4})\theta^4$, we have $t(k_x, k_y) = (C_{p4} - C_{s4})\frac{\sin 2\varphi}{2}\theta^4$, and then forth-order differential and third-

order differential derived by forth-order differentiation can be realized.

According to the transmission band diagram, the difference $t_p - t_s$ varies continuously from $t_p - t_s \propto \theta^2$ to

$t_p - t_s \propto \theta^6$ with the change of the incident wavelength (there is no odd-power because the one-dimensional photonic crystal is isotropic in the horizontal direction), which makes the second-order, fourth-order and even sixth-order differential be displayed in turn, and the first-, third- and fifth-order differential derived from them are also found.

(4) The authors claim that “Third- and fourth-order differentiations that have never been realized before.” I suggest the authors add some discussions to clarify why it is important to realize the higher-order differentiations.

Response: Thank you very much for your valuable suggestion, which is helpful to improve our article! We have supplemented in the discussion section.

Higher-order spatial differentiation of the change position of the samples can obtain finer and sharper peaks. As said by Professor Shanhui Fan of Sandford University, “First-order differentiation, which is less effective for image sharpening compared to higher-order derivatives” in one of his published paper (*Optics Letters* **46**, 13 (2021)). It shows that higher-order differentiation plays an important role in image sharpening.

Moreover, it can be anticipated that abundant orders spatial differentiations can be used together to solve the differential equations (*Sci. Rep.* **9**, 12928 (2019); *J. Opt.* **18**, 075102 (2016); *Nature Reviews Materials* **6**, 207–225 (2021)).

4. In addition, the authors should check the manuscript carefully. For example, figure 3b is not clearly epically the areas of the edges, the authors should provide high resolution image. The scare bar on Figure 6 is missing.

Response: Thank you very much for your valuable suggestion. We have inserted the enlarged view of edges on figure 3 and inserted the scale bar on figure 6.

Reviewer #3 (Remarks to the Author):

The authors propose multiple-order analog spatial differentiator with simple multilayer structures. By designing the spectrum transfer function of a single device, different order spatial differentiations are realized at different incident angles and different wavelength. I think that it is quite interesting that a simple structure can be used to perform the optical computation of multiple-order differentiation. As the experimental demonstrations, such a device is also useful for the image processing and surely other applications.

Response: We really appreciate these positive comments from the reviewer.

Below there are some of my comments for the authors' consideration.

- What resolution for such high-order differentiation, the three- and four-order differentiation? Any advantages over the first- or the second-order differentiation?

Response: Thank you for raising these insightful comments.

Thank you for reminding us the resolution of multi-order differentiations. It is of great importance to improve the quality of our manuscript. We specially added a section on the resolution "Section 5: Resolution of multi-order differentiation" in the supporting information. In this section, we used a rectangle input field to represent the information of USAF test target, of which the output fields after the differential operations of different orders were analyzed. Then, the resolution of multi-order spatial differentiation was obtained according to Rayleigh criterion.

As said by Professor Shanhui Fan of Sandford University, higher-order differential can bring sharper peaks at the edges of the image, that means higher-order differentiation plays an important role in image sharpening(*Optics Letters* **46**, 13 (2021)). Moreover, solving the differential equations with constant coefficients can be realized owing to the use of abundant differential operators (*Sci. Rep.* **9**, 12928 (2019); *J. Opt.* **18**, 075102 (2016); *Nature Reviews Materials* **6**, 207–225 (2021)).

Since the fundamental mode Gaussian beam and Hermite-Gaussian beam are solved by the same Helmholtz equation, Hermite-Gaussian beam can be derived from fundamental mode Gaussian beam. By combining different order partial differentiations, we can get various Hermite-Gaussian beams from fundamental mode Gaussian beams. Therefore, beam shaping is also one of the functions of the proposed multi-order differential chip.

In the section discussion of the main text, we have added some descriptions on the value of multi-order spatial differentiation. Thank you again.

- Since the different polarizations of the incident and the output light are analyzed to realize the differentiation, does the spin Hall effect of light also contribute to the spatial differentiation, such as *Physical Review Applied* **11**, 034043 (2019) and *Advanced Photonics* **2**, 016001 (2020)?

Response: Thank you for your valuable comment. The two literatures are very interesting, which expand our

understanding of spatial differentiation. As far as we know, a spatial differentiator based on the SPP was proposed in 2017, which has been widely read and cited by researchers (*Nature Communications* **8**, 15391 (2017)). We have cited this NC paper in the Introduction of our manuscript. Due to the transverse-magnetic (TM) polarization property of the SPP, only TM-polarized light can be modulated, thus this NC paper is limited in the scope of scalar optical field.

The two literatures mentioned here extended spatial differentiation to vector field, and it can be seen as a general result regardless of material composition or incident angles, which is a huge advancement over the previous work. In the two literatures, the spin Hall effect was used to make the reflected light shearing interfered, and resulted in the spatial differentiation. Focused light was introduced here to provide polarization in a direction perpendicular to the polarization of the incident light through the reflection or refraction of the interface.

In our manuscript, t_p and t_s were modulated to obtain the desired OTE, thereby the multi-order differentiations could be implemented. Here we call this method “transmissivity modulation method”. It is worth noting that small changes in t_p and t_s will have a huge impact on the order of differential. In the works published in *Physical Review Applied* and *Advanced Photonics*, the spin angular momentum and the orbital angular momentum of the LCP and RCP were converted at the interface to make them propagate in different directions, thus realizing the first-order differentiation along the direction perpendicular to the incident plane. The spin Hall effect can occur at any interface, and the difference between the transmissivity or reflectivity of the interface for s- and p-polarization which plays an important role in differential order of the transmission modulation method, only affects the angle of separation of LCP and RCP, and does not affect the order of differentiation.

Thus, the transmission modulation method and the SHE based method are essentially different. When the first- and third-order differentiations are realized in our work, the differential effects generated by the transmission modulation method and the SHE based method will coexist. But the direction deflection between LCP and RCP is small due to the small incident angle (NA of incident light is less than 0.2) that makes the differential effect generated by spin Hall effect weak. Since the axis of the light generating the second- and fourth-order differentiation is perpendicular to the optical chip, the homogeneity in the horizontal direction will offset the spin Hall effect.

The main differences of our work from the published literature are that multiple-order analog spatial differentiator can be realized with one optical element, and the proposed planar chip works in the transmission mode, which is the operation mode of most commercial microscopes, so it is easy to adopt the chip in the commercial imaging system.

The two literatures are very valuable in enlightening readers and helping us understand the leading-edge progress, so we have cited them in the revised manuscript.

- Any discussion or general suggestion to design the structure for the other operation wavelength?

Response: Thank you for your reminder and suggestion. We have added a Section in the supporting information to discuss the topic of wavelength customization. For your convenience, we copy these descriptions here.

“Section 6: Method of wavelength customization

In the previous section, we discussed the differential case at specific wavelengths, and then a question will be raised: can we customize the operation wavelength to obtain a chip that works at the desired wavelength?

In order to answer this question, we sorted out the process of chip design. Firstly, two materials with known refractive index were selected as the basic periodic structures of one-dimensional photonic crystals (the proposed planar photonic chip). As the widely used materials, Si_3N_4 and SiO_2 were selected due to the low cost and mature processing technology for thin films deposition. Secondly, a total number of 40 layers of this chip is designed to realize a large band gap depth (**Fig. 1b** and **Fig. S8c**), so that the form of the OTF changes rapidly with the wavelength to achieve the conversion of multi-order differentiations. Thirdly, the layer thicknesses of the two dielectric layers needs to be determined. By adjusting the thicknesses of the two dielectric layers, the photonic band structure will move along the direction of increasing or decreasing incident light frequency. In this way, we can tune the working frequency (or wavelength), at which the spatial differentiations can be realized. As a result, the operation wavelength of the photonic chip can be customized (**Fig. S7**).”

List of Changes

1. As suggested by Reviewer 1 and 3, we have added “**Section 5: Resolution of multi-order differentiation**” and **Figure S6** in the **supporting information** and the text “**Detailed analysis on the resolution of the planar chip based differential system is given in Section 5 and Fig. S6 of the supplementary information.**” in **line 182 of the main text**.
2. To add Figure S6 in the supporting information, **the original Figure S6 was changed to Figure S7 and the original Figure S7 was changed to Figure S8** in the **main text** and the **supporting information**.
3. As suggested by Reviewer 1, we have cited references in **line 34** (*ACS Photonics* **7**, 1799-1805 (2020); *Optics Express* **30**, 17806-17823 (2022)) and in **line 41** (*Optics Letters* **45**, 6867-6870 (2020), *Optics Express* **27**, 27295-27307 (2019)) of the main text.
4. As suggested by Reviewer 2 and 3, we have added a **description of the importance of higher-order differential** in the discussion section and inserted references in **line 297** (*Nature Reviews Materials* **6**, 207–225 (2021), *Journal of Optics* **18**, 6 (2016), *Scientific Reports* **9**, 10 (2019)), **line 298** (*Physical Review Applied* **11**, 034043 (2019), *Optics Express* **22**, 25084-25092 (2014), *Optics Letters* **39**, 1278-1281 (2014)) and **line 300** (*Optics Letters* **46**, 3247-3250 (2021)) of the main text.
5. As suggested by Reviewer 2, we have inserted the **magnified images of edges** in **figure 3** and a text “**The insets in (d), (e), (f), and (g) show magnified images of the regions marked using a line-box in each case.**” was added in **line 544 of main text**.
6. As suggested by Reviewer 2, we have inserted the **scale bar** in **figure 6**.
7. We cited a paper on spin Hall effect in **line 34** (*Physical Review Applied* **11**, 034043 (2019) and *Advanced Photonics* **2**, 016001 (2020)) of the main text and a paper on phase mining in **line 233** (*Advanced Photonics* **2**, 016001 (2020)) of the main text.
8. As suggested by Reviewer 3, we have added “**Section 6: Method of wavelength customization**” in the **supporting information** and we inserted the text “**(More details are given in Section 6 of the supplementary information)**” in **line 223 of the main text**.
9. In **line 180 of the main text**, we have removed space between “Fig. S5” and “a”.

REVIEWER COMMENTS

Reviewer #2 (Remarks to the Author):

In the revised version of the manuscript, the authors have done a commendable job at shedding lights upon the questions raised during the first review process. This not only makes the manuscript more insightful, but also makes the whole experimental procedure easier to follow. However, a minor issue remains.

I agree the authors' opinion that this name “photonic chip” can attract the more attention from a wider audience of the journal Nature Communications. However, I still think the name is not strictly accurate. I think that the Chip-scale differentiator maybe is more appropriate. I would like leave it to the authors to decide whether make this change.

Reviewer #3 (Remarks to the Author):

The authors revised the manuscript in accordance with the Reviewers' comments, and I would like to recommend to publish it as present.

A Point-by-point response to the referees' comments

REVIEWERS' COMMENTS

Reviewer #2 (Remarks to the Author):

In the revised version of the manuscript, the authors have done a commendable job at shedding lights upon the questions raised during the first review process. This not only makes the manuscript more insightful, but also makes the whole experimental procedure easier to follow. However, a minor issue remains.

I agree the authors' opinion that this name “photonic chip” can attract the more attention from a wider audience of the journal Nature Communications. However, I still think the name is not strictly accurate. I think that the Chip-scale differentiator maybe is more appropriate. I would like leave it to the authors to decide whether make this change.

Response: Thank you very much for the positive comments.

We fully understand the concerns on the term “photonic chip” from the respected reviewer. After we received this comment, we have gone through many references in this field.

In our opinions, we really think the term “photonic chip” is more suitable for our work. We are afraid that the term “chip-scale differentiator” will induce some confusions from the broad readers. They will ask what the term “chip-scale” means.

On the other hand, we have found similar title from the journal Nature Photonics, such as the following references.

- (1) **Chip-based** wide field-of-view nanoscopy, Nature Photonics 11, 322–328 (2017)
- (2) Structured illumination microscopy using a **photonic chip**, Nature Photonics 14, 431–438 (2020)
- (3) Dark field on a **chip**, Nature Photonics 14, 266–267 (2020)

They also named the substrate as the photonic chip. The situation is the same as our work. As a result, we prefer to use the term “photonic chip” in our manuscript.

We really appreciate the kind comments and suggestions from the respected reviewers. Thank you again.

Reviewer #3 (Remarks to the Author):

The authors revised the manuscript in accordance with the Reviewers' comments, and I would like to recommend to publish it as present.

Response: Thank you very much for the positive comments. We really appreciate the valuable comments from the reviewers, which significantly help to improve the quality of our manuscript.

List of Changes

1. We changed all '(a. u.)' to '(arb. units)' in figures.
2. Descriptions of abbreviations were added to legends of Figure 1 and Figure 3.

-
3. The texts in Figure 2c, 3a and S8a were changed from 'potonic chip' to 'photonic chip'.
 4. The insets showing magnified images of the regions marked using line-boxes were inserted in Figure 4a and 4b.
 5. The order of the sections in the main article was adjusted to meet the requirements.
 6. The Methods section was renamed as 'Methods'.
 7. The horizontal ordinate label of Figure S1b was changed from ' k_x / k_0 ' to ' k_r / k_0 '
 8. The vertical ordinate label of Figure S3c was changed from ' $t(k_p)$ ' to ' $t(k_r)$ ' and the ordinate label of Figure S3f was changed from ' $|t(k_p)|$ ' to ' $|t(k_r)|$ '.
 9. We typeset vectors in bold without italics.
 10. We typeset the text 'Fig.', text 'Eq.' and label of the formulas in supporting information as not bold.
 11. We refrained from using words such as 'new/novel/first', when referring to the scientific findings, and also removed exaggerated language such as 'extremely/outstanding'.